# Inter- and intra-rater reproducibility of quantitative T1 measurement using semiautomatic region of interest placement in myometrium

**Sadahiro Nakagawa**[1], **Takahiro Uno**[1], **Shunta Ishitoya**[2], **Eriko Takabayashi**[2], **Akiko Oya**[2], **Wakako Kubota**[2], **Atsutaka Okizaki**[2] *

**1** Division of Radiology, Asahikawa Medical University Hospital, Asahikawa, Japan, **2** Department of Radiology, Asahikawa Medical University, Asahikawa, Japan

* okizaki@asahikawa-med.ac.jp

## Abstract

### Purpose

This study aimed to investigate the inter- and intraobserver reproducibility of quantitative T1 (qT1) measurements using manual and semiautomatic region of interest (ROI) placements. We hypothesized the usefulness of the semiautomatic method, which utilizes a three-dimensional (3D) anatomical relationship between the myometrium and other tissues, for minimizing ROI placement variation, thereby improving qT1 reproducibility compared to the manual approach. The semiautomatic approach, which considered anatomical relationships, was expected to enhance reproducibility by reducing ROI placement variabilities.

### Materials and methods

This study recruited 23 healthy female volunteers. Data with variable flip angle (VFA) and inversion recovery were acquired using 3D-spoiled gradient echo and spin echo sequences, respectively. T1 maps were generated with VFA. Manual and semiautomatic ROI placements were independently conducted. Mean qT1 values were calculated from the T1 maps using the corresponding pixel values of the myometrial ROI. Inter- and intraobserver reproducibility of qT1 values was investigated. The inter- and intraobserver reproducibility of qT1 values was evaluated by calculating the coefficient of variation (CoV). Further, reproducibility was evaluated with inter- and intraobserver errors and intraclass correlation coefficients (ICCs). Bland–Altman analysis was utilized to compare the results, estimate bias, and determine the limits of agreement.

### Results

The mean inter- and intraobserver CoV of the qT1 values for semiautomatic ROI placement was significantly lower than those for manual ROI placement ($p < 0.05$ and $p < 0.01$, respectively). ICCs for semiautomatic ROI placement were greater than those for manual ROI placement. Further, the mean inter- and intraobserver errors for semiautomatic ROI

**Data Availability Statement:** The minimal dataset underlying the results described in this study are within the manuscript and its supporting information files. The source code analysis of

Microsoft Visual Studio 2010 image analysis code and MATLAB, along with an example image that supports the findings of this study, has been openly published on Zenodo at [https://zenodo.org/deposit?page=1&size=20]. Additionally, any other data required to fully replicate the reported study findings can be obtained by contacting the corresponding author upon reasonable request.

**Funding:** Atsutaka Okizaki received Research funding of Nihon Medi-physics Co. and FUJIFILM Toyama Chemical Co., Ltd. The funders had no role in study design, data collection and analysis, decision to publish, or preparation of the manuscript.

**Competing interests:** The other authors have no conflict of interest. This does not alter our adherence to PLOS ONE policies on sharing data and materials.

placement were significantly lower than those for manual ROI placement ($p < 0.05$ and $p < 0.01$, respectively).

## Conclusion

Semiautomatic ROI placement demonstrated high reproducibility of qT1 measurements compared with manual methods. Semiautomatic ROI placement may be useful for evaluating uterine qT1 with high reproducibility.

## Introduction

Magnetic resonance imaging (MRI) data quantitative image analysis may be useful for comparing subjects. Quantitative T1 (qT1), T2, and apparent diffusion coefficient values have been utilized to investigate various diseases [1–4]. Quantitative image analysis has the advantage of better tissue identification and classification compared with the visual assessment of traditional T1- and T2-weighted images [4–6]. QT1 values have been extensively used to study the central nervous system over the past several decades [7–10].

Davenport et al. revealed qT1, using dynamic contrast-enhanced MRI with variable flip angle (VFA) in patients with uterine myoma, to be crucial for characterizing the lesion and monitoring drug effects [11]. Dominant leiomyoma qT1 measures were used to predict uterine and dominant leiomyoma size reductions after uterine artery embolization (UAE) [12].

QT1 values vary based on various factors, including magnetic field strength, imaging sequence, and patient characteristics [13,14]. Therefore, each institution must define reference values for standardization. In principle, these reference values would require high reproducibility for quantitative measurements. QT1 values sometimes vary, based on the location, shape, and area of the region of interest (ROI) when the ROI is manually set. Heye et al. revealed that semiautomatic ROI placement enhanced the reproducibility of pharmacokinetic parameters; however, the method is only available using dynamic enhanced MRI [15]. Chapiro et al. emphasized the usefulness of semiautomated three-dimensional (3D) tumor segmentation but noted the necessity of acquiring contrast-enhanced T1-weighted images [16]. Highly reproducible ROI placement without the need for a contrast agent is preferable for quantitative MRI.

Therefore, we developed a semiautomatic myometrial ROI placement method without a contrast agent. We hypothesized that a semiautomatic method, using a 3D anatomical relationship between the myometrium and endometrium, may be useful for minimizing variation by manual ROI placements and improving ROI placement reproducibility. This study aimed to investigate the inter- and intraobserver reproducibility of qT1 measurements using manual and semiautomatic ROI placements.

## Materials and methods

### Subjects

Our institutional research ethics committee at Asahikawa Medical University approved the study protocol following the principles of the Declaration of Helsinki. All subjects were informed regarding the content of the study and signed written informed consent. From May 2018 to April 2019, 56 subjects underwent pelvic MR imaging, incorporating VFA for qT1 measurements in our hospital.

This study excluded subjects who met the following exclusion criteria: (a) subjects who presented with uterine mass lesions (n = 5), (b) subjects who had misregister between inversion recovery (IR) and each VFA image (n = 13), and (c) subjects who observed motion artifacts in the spoiled gradient recalled echo (SPGR) image acquired for qT1 with VFA (n = 15). Our study enrolled 23 healthy women volunteers (age range: 22–51 years; mean age ± standard deviation [SD]: 33 ± 11 years).

## Pelvic MRI

All pelvic MRI data were acquired using a 3-T scanner (Discovery MR750w; GE Healthcare, Waukesha, WI, USA). Two radiologists with 18 and 3 years of radiological experience, respectively, performed visual examinations to confirm that the subjects did not have mass lesions in the pelvic MRIs (T2-weighted image, T1-weighted image, and diffusion-weighted image). Any disagreements in interpretation were resolved by consensus following the discussion. Radiologists diagnosed all enrolled subjects as normal.

## Magnetic resonance pulse sequences

Table 1 summarizes the MR pulse sequences used for the IR and VFA qT1 measurements.

For VFA, 3D-SPGR was conducted with B1 field inhomogeneity correction using the Bloch–Siegert method [17]. T1 maps were generated pixel by pixel using MATLAB (Math-Works, Natick, MA) with two flip angles (FAs) of 4˚ and 18˚ (S1 Appendix) [3]. The code used to develop the T1 map was made available through Zenodo, DOI: 10.5281/zenodo.7808462 (https://doi.org/10.5281/zenodo.7808462). The Bloch—Siegert method was used to rapidly acquire a B1 map with an acquisition time of 14 s for B1 field inhomogeneity correction [18]. The scan parameters were as follows: repetition time: 6 ms; echo time: 4 ms; FA: 5˚; FOV: $53 \times 53$ cm$^2$; slice thickness: 10 mm; and matrix size: $64 \times 64$. We acquired oblique axial sections of approximately proximal one-third of the uterine body using spin echo sequences because the physiological movement of adjacent organs is less likely to affect this section [19], thereby minimizing motion artifact effects. We used an inversion time of 200 ms for visual assessment of the highest contrast image between the signal intensity of the myometrium and endometrium. IR was used to adjust the center of the MR image with SPGR.

## ROI placement for the qT1 measurements

Two expert MRI technicians (observers 1 and 2), with 6 and 14 years of experience in pelvic MRI, respectively, evaluated the MR images using IR images and two SPGRs to confirm image quality and image registration. Observers 1 and 2 independently performed manual and semi-automatic ROI placements without consultation.

## Manual ROI placement

Manual ROI placement was performed using MATLAB (S2 Appendix) using the following procedure. (1) The T1 map data was transferred to the MATLAB workspace. (2) The observer selected an IR image in the MATLAB application to place the ROI. (3) The provided code was executed to open the image segmentation application in MATLAB. (4) A whole uterine ROI was manually placed to trace the external contour of the myometrium. (5) An endometrial ROI was manually placed to trace the external contour of the endometrium. (6) The myometrial ROI was developed by subtracting the endometrial ROI from the whole uterine ROI. (7) The mean qT1 values were calculated from the corresponding pixel values of the T1 map

**Table 1. MR pulse sequences.**

| Parameter | IR<br>TSE | VFA for qT1 measurements<br>3D-SPGR |
|---|---|---|
| Acquisition plane | Oblique | Oblique |
| TR (ms) | 7800 | 5.0–5.6 |
| TE (ms) | 10 | 2.5 |
| FA (°) | 180 | 4 and 18 |
| FOV (cm) | 25 | 25 |
| NEX | 1 | 1 |
| Matrix (Frequency) | 256 | 256 |
| Matrix (Phase) | 256 | 256 |
| TI (ms) | 200 | - |

MR: Magnetic resonance; IR: Inversion recovery; TSE: Turbo spin echo; VFA: Variable flip angle; qT1: Quantitative T1; 3D-SPGR: 3D-spoiled gradient recalled echo; TR: Repetition time; TE: Echo time; FA: Flip angle; FOV: Field of view; NEX: Number of excitations; TI: Inversion time.

within the myometrial ROI. The code for manual ROI placement was made available through Zenodo, DOI: DOI/10.5281/zenodo.7811343 (https://doi.org/10.5281/zenodo.7811343).

The endometrial ROI was then drawn over the endometrium (Fig 1B). The myometrial ROI was defined by subtracting the endometrial ROI from the whole uterine ROI (Fig 1C). The myometrial region of interest included the junctional zone. The mean qT1 values were calculated from the corresponding pixel values of the T1 map within the myometrial ROI.

## Semiautomatic ROI placement

We developed in-house software using Microsoft Visual Studio 2010 (Microsoft Corporation, Redmond, WA, USA) to read and display digital imaging and communications in medicine (DICOM) files on a personal computer with Windows 10, with a 64-bit operating system (Microsoft). We converted the DICOM images to Joint Photographic Experts Group (JPEG) format using MATLAB before analysis (S3 Appendix). The source code for the in-house software for reading and displaying DICOM files, as well as the MATLAB code for converting DICOM images to JPEG format, has been made available through Zenodo, DOI: 10.5281/zenodo.7807266 (https://doi.org/10.5281/zenodo.7807266).

The following semiautomatic procedures were used to place myometrial ROI (Fig 1D–1F, S4 Appendix). (1) The user determined the center of the endometrium. (2) The user traced the external contour of the myometrium (Fig 1D). (3) The software automatically extracted the endometrial ROI using a signal intensity threshold (Fig 1E), which was initially set within ±3% of the endometrial signal intensity determined by the user and was increased in increments of 1% to maximize the area of the endometrial ROI, while excluding the external contour of the myometrium. (4) Pixel values were sampled from the area between the endometrial ROI and the external contour of the myometrium. (5) The mean and SD of the sampled pixel values were calculated. (6) The myometrial ROI was automatically extracted based on pixels that were within ±1.96 SD of the mean signal intensity. (7) The myometrial ROI was placed on the SPGR image. (8) We calculated the mean pixel value of the myometrial ROI. (9) The sum of squared deviation was calculated with ROI location variation, and the vertical and horizontal coordinates were changed from −10 to +10 pixels. (10) The optimal combination of vertical and horizontal coordinates with a minimum sum of squared deviation was obtained, and experts visually confirmed the appropriateness of the ROI location. (11) The mean qT1 value was calculated from the corresponding pixel values of the T1 map within the optimally placed myometrial ROI (Fig 1F). Eventually, the required operations by the user were only two steps:

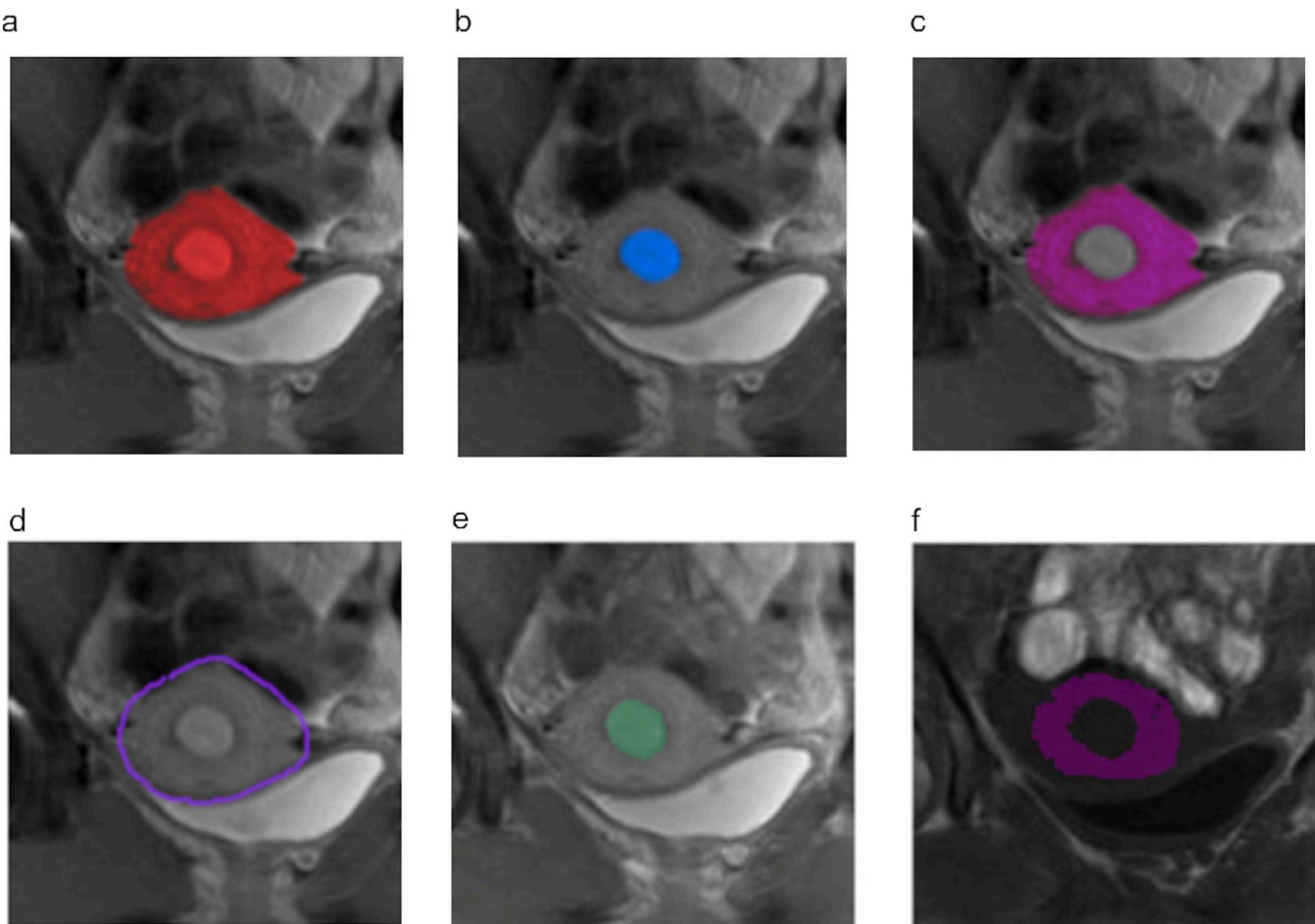

**Fig 1.** Examples of ROI definition with manual placement (a-c) and semiautomatic placement (d-f) in IR and VFA. (a) Whole uterine ROI (red area), (b) endometrial ROI (blue area), (c) myometrial ROI (purple area), (d) External contour of the myometrium (purple line), and (e) endometrial ROI (green area) in IR and (f) myometrial ROI (purple area) in VFA. Several pixels located inside the myometrium were not identified as myometrium with semiautomatic ROI placement (f). Those pixels with more than the mean of +1.96 SD of myometrial signal intensity, may be caused by blood vessel flow.

(1) click the center of the endometrium and (2) roughly draw the shape of the myometrium manually. Our algorithm does not use any active contour model for ROI extraction. The Code for semiautomatic ROI placement was made available through Zenodo, DOI: DOI/10.5281/zenodo.7805680 (https://doi.org/10.5281/zenodo.7805680).

## Reproducibility

Observers 1 and 2 independently evaluated manual and semiautomatic ROI placements to assess interobserver reproducibility. Observer 1 repeated the measurement within 2 weeks of the first measurement to assess intraobserver reproducibility. The supporting information (S1–S4 Files) included the datasets for the evaluation of inter- and intraobserver reproducibility in T1 values, conducted through both manual and semiautomatic ROI placement by two observers.

## Statistical analysis

The inter- and intraobserver reproducibility of qT1 values was assessed by calculating the coefficient of variation (CoV). Further, reproducibility was evaluated using inter- and intraobserver errors and intraclass correlation coefficients (ICCs).

Inter- and intraobserver errors were calculated as follows:

$$(\text{Interobserver error}) = \frac{|\text{qT1 by observer 1} - \text{qT1 by observer 2}|}{(\text{qT1 by observer 1} + \text{qT1 by observer 2})/2} \times 100\ (\%),$$

$$(\text{Intraobserver error}) = \frac{|\text{qT1}_{1st} - \text{qT1}_{2nd}|}{(\text{qT1}_{1st} + \text{qT1}_{2nd})/2} \times 100\ (\%),$$

where $\text{qT1}_{1st}$ and $\text{qT1}_{2nd}$ are the first and second measurements of qT1 by observer 1, respectively. The Wilcoxon test and Bland–Altman analysis were used to compare the results, estimate bias, and determine the limits of agreement. A $p$-value of $<0.05$ was considered statistically significant. Bland–Altman analysis was used to assess the proportional bias. All statistical analyses were performed using Statistical Package for the Social Sciences software version 19.0 (SPSS-IBM, Armonk, NY). G*power was used for analysis for calculating the sample size. Using the usual significance level of 0.05 and a power of 0.80 for statistics by Wilcoxon signed-rank test, a sample size of a minimum of 20 subjects would be required to identify the difference in the mean error between the two groups. We measured the qT1 evaluation time with manual and semiautomatic ROI placements, and a paired $t$-test or Wilcoxon signed-rank test was used to compare the two methods depending on whether or not the distribution was normal.

## Results

The mean CoVs of qT1 values for semiautomatic ROI placement by observers 1 and 2 were significantly lower than those for manual ROI placement ($p < 0.05$; Fig 2A).

Furthermore, both the first and second measurements of semiautomatic ROI placements by observer 1 were significantly lower than those of manual ROI placements ($p < 0.01$; Fig 2B).

The mean inter- and intraobserver errors for semiautomatic ROI placement were significantly lower than those for manual ROI placement ($p < 0.05$ and $p < 0.01$, respectively; Fig 2C and 2D).

Table 2 summarizes the ICCs.

The inter- and intraobserver reproducibility of semiautomatic ROI placement was excellent at $>0.98$ (Table 2; $p < 0.01$). The 95% confidence intervals for the ICC did not overlap between manual and semiautomatic ROI placements in both interobserver and intraobserver reproducibility. The ICCs for semiautomatic ROI placement were higher than those for manual ROI placement.

Fig 3 shows the Bland–Altman plots. Any combination demonstrated no fixed bias (Fig 3A–3D).

Positive proportional bias emerged in the qT1 measurements between observers 1 and 2 using semiautomatic ROI placement (R = 0.461, $p = 0.027$; Fig 3B). A negative proportional bias was determined between the first and second qT1 measurements using manual ROI placement (R = 0.532, $p = 0.009$; Fig 3C). However, no proportional bias was observed in the measurements between observers 1 and 2 using manual ROI placement (R = 0.224, $p = 0.304$; Fig 3A). Similarly, no proportional bias was observed between the first and second measurements using semiautomatic ROI placement (R = 0.11, $p = 0.959$; Fig 3D). The means and SDs of time with manual and semiautomatic ROI placements were 55 ± 4 s and 44 ± 5 s, respectively. The times with semiautomatic ROI placement were significantly smaller than those with manual ROI placement ($p < 0.001$).

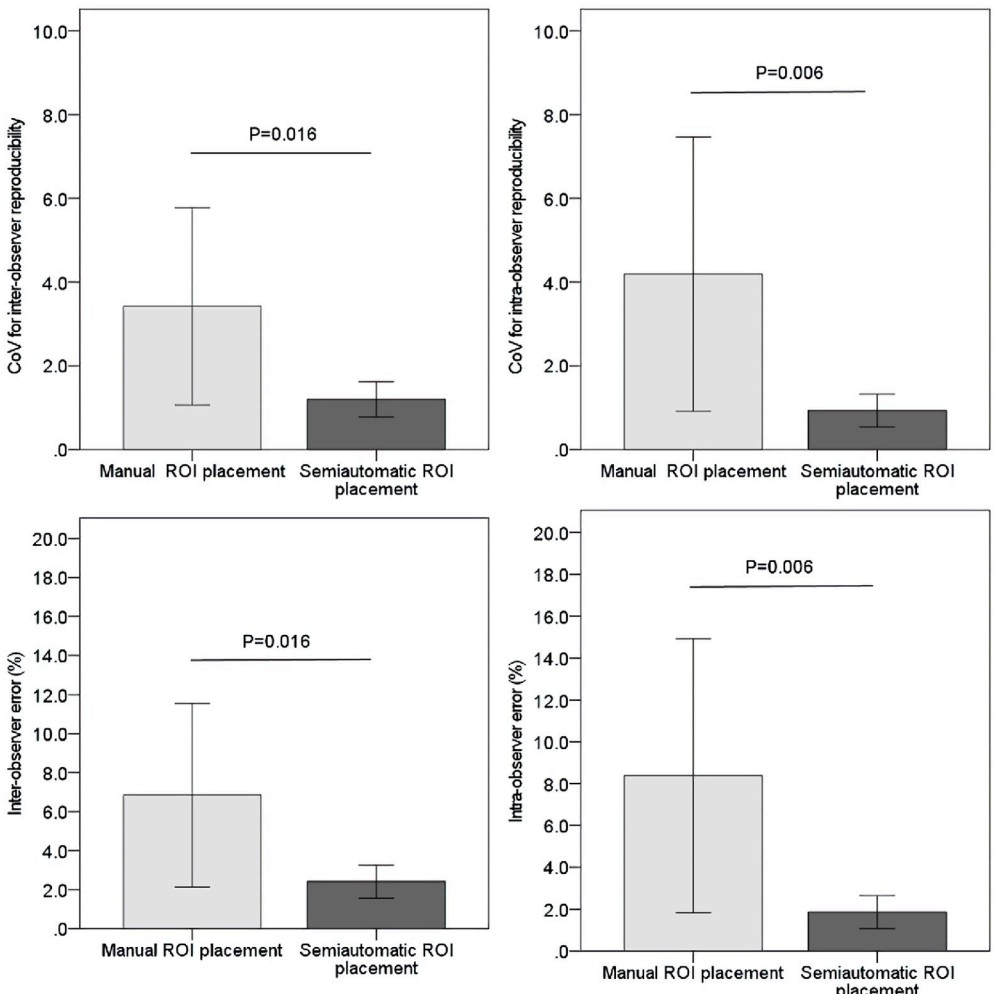

**Fig 2. Inter- and intraobserver reproducibility of qT1 values by calculating the coefficient of variation (CoV) and inter- and intraobserver errors.** (a) CoV for interobserver reproducibility, (b) CoV for intraobserver reproducibility, (c) interobserver error, and (d) intraobserver error. Error bars show the mean ± standard error.

## Discussion

This study revealed good reproducibility for both manual and semiautomatic ROI placements. Further, the CoV and inter- and intraobserver errors for semiautomatic ROI placement were significantly lower than those for manual ROI placement.

Highly reproducible ROI placements are advantageous for quantitative image analysis. Heye et al. reported that semiautomatic lesion segmentation significantly reduced interobserver variability compared with the manual approach, but not intraobserver variability using

**Table 2. Reproducibility (ICC).**

| Average ICC (95% CI) | Interobserver reproducibility | Intraobserver reproducibility |
|---|---|---|
| Manual ROI placement | 0.807 (0.601–0.913) * | 0.794 (0.578–0.906) * |
| Semiautomatic ROI placement | 0.982 (0.958–0.992) * | 0.988 (0.971–0.985) * |

ICC: Intraclass correlation coefficient; 95% CI: 95% confidence interval; ROI: Region of interest, * $p < 0.01$.

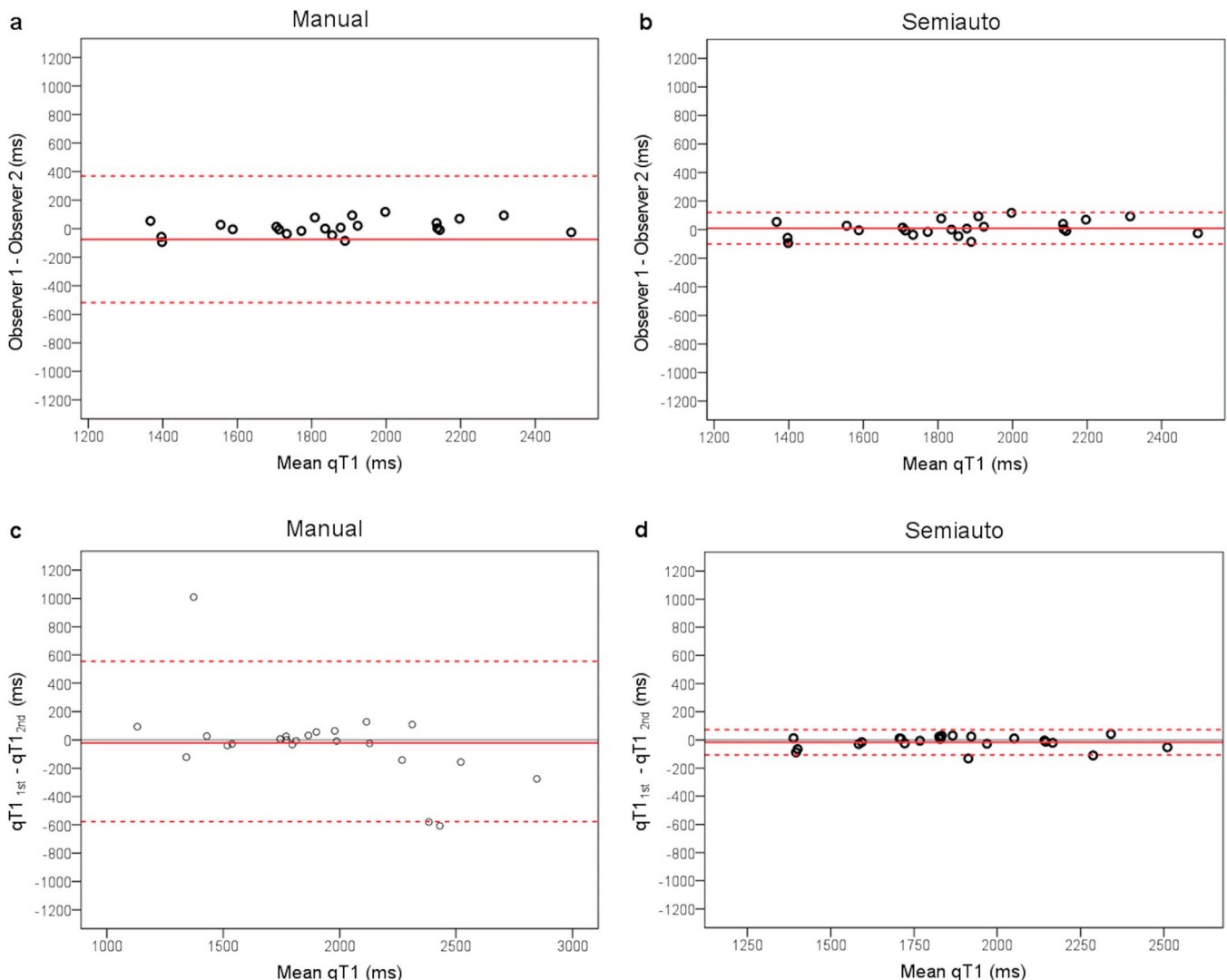

**Fig 3. Bland–Altman analyses estimated the bias and limits of agreement.** Bland–Altman analysis of qT1 measurement by observers 1 and 2 with manual (a) and semiautomatic ROI placements (b) Analysis of the first and second measurements with manual (c) and semiautomatic ROI placements (d) The solid and dotted red lines indicate the mean differences and 95% limits of agreement, respectively. Mean differences were −72 (a), 11 (b), −20 (c), and −17 ms (d), and the 95% limits of agreement were 370 and −518 ms (a), 122 and 99 ms (b), 556 and 596 ms (c), and 73 and 107 ms (d).

the semiautomatic approach [15]. In contrast, our study revealed significant improvements using the semiautomatic algorithm for both inter- and intraobserver errors. Moreover, a contrast agent was not necessary for semiautomatic ROI placement.

The Bland–Altman analysis detected a negative proportional bias of intraobserver variation for manual ROI placement. This bias may be associated with two data points, as shown in the lower right of Fig 3C. The qT1 values differed between the myometrium and areas around the myometrium, which were derived from structures outside the uterus, such as the bowel, vessels, and urine in the bladder. Therefore, the myometrial ROI may have included surrounding tissues when using manual ROI placement. Such errors are less likely to occur when using the algorithm for semiautomatic ROI placement.

The algorithm for semiautomatic ROI placement demonstrated high reproducibility. In principle, the algorithm can be applied to other medical imaging modalities for delineating the

uterus because it uses pixel values that correspond to anatomical structures. This software was specifically designed for uterine analysis, but its programmatic design also theoretically allows for its use in analyzing other organs. With further research, the use of this software may extend beyond its current scope. Therefore, we plan to continue investigating and developing this software to explore its potential applications. Absolute native T1 values vary depending on MRI field strength, acquisition method, and vendor. Thus, the results of this study cannot be directly compared with qT1 value evaluation from another institution [13]. However, after achieving standardization using a certain method, comparisons of absolute qT1 may be possible. The subjects in our study were healthy volunteers, but our approach may apply to abnormal myometrium cases. We plan to apply our method to patients with uterine tumors, such as endometrial cancer, cervical cancer, and fibroid.

The degree of reproducibility for manual ROI placement in our study was not as high as that previously reported [11,12]. Therefore, our comparisons of reproducibility between manual and semiautomatic ROI placements are reliable.

Several studies reported on the use of uterus T1 mapping for evaluating fibroid and healthy volunteers. Sipola et al. reported that qT1 in patients with leiomyoma predicted treatment effects after uterine artery embolization [12]. Suomi et al. revealed that heating efficacy in high-intensity focused ultrasound therapy was estimated using qT1 [20]. Takatsu et al. evaluated the qT1 of normal female reproductive organs in the luteal phase of the menstrual cycle [21].

This study has several limitations. First, this is a single-center study. A multicenter randomized trial is required to confirm the use of our semiautomatic ROI placement algorithm. Second, we included a relatively small sample size (n = 23). We found statistically significant differences in CoV and inter- and intraobserver errors between manual and semiautomatic ROI placements but with no significant differences in ICCs. However, the ICCs for semiautomatic ROI placement may be greater than those for manual ROI placement in a larger sample. Third, our study included healthy volunteers. In principle, the algorithm should apply to uterine diseases because the average qT1 values for abnormal lesions differ in the myometrium and other organs. However, further studies are required to evaluate its application in patients with uterine diseases. Fourth, this study used a short repetition time (5.0–5.6 ms) because of the limited scan time in clinical practice; however, the CoV values of our data were equivalent to those in previous reports [11,12]. Fifth, the range of qT1 with VFA in the myometrium was quite wide, ranging approximately from 1200 ms to 2500 ms. This may be because the subjects were composed of postmenopausal menstrual women, and their menstrual cycle would have varied. Sixth, an analysis of the relationship between qT1 and the menstrual cycle was not investigated.

In conclusion, the semiautomatic ROI placement algorithm demonstrated high reproducibility of qT1 measurements compared with manual methods. Semiautomatic ROI placement may be useful for evaluating uterine qT1 with high reproducibility.

## Supporting information

**S1 Appendix. User guide for T1 map analysis using MATLAB code from "Inter- and Intra-rater reproducibility of quantitative T1 measurement using semiautomatic ROI placement in myometrium" Research Paper on Zenodo.** https://doi.org/10.5281/zenodo.7808462. (DOCX)

**S2 Appendix. User guide for manual ROI placement analysis using MATLAB code from "Inter- and Intra-rater reproducibility of quantitative T1 measurement using semiautomatic ROI placement in myometrium" research paper on Zenodo.** https://doi.org/10.5281/

zenodo.7811343.
(DOCX)

**S3 Appendix. User guide for converting DICOM to JPEG using MATLAB from the research paper "Inter- and Intra-rater reproducibility of quantitative T1 measurement using semiautomatic ROI placement in myometrium" is possible with the MATLAB procedure and test data provided in the corresponding repository on Zenodo.** https://doi.org/10.5281/zenodo.7807266.
(DOCX)

**S4 Appendix. User guide for semiautomatic ROI placement analysis using Microsoft Visual Studio 2010 code from "Inter- and Intra-rater reproducibility of quantitative T1 measurement using semiautomatic ROI placement in myometrium" research paper on Zenodo.** https://doi.org/10.5281/zenodo.7805680.
(DOCX)

**S1 File. T1 values for interobserver reproducibility assessment; Excel data with manual ROI placement by observer 1 and 2.**
(XLSX)

**S2 File. T1 values for intraobserver reproducibility assessment; Excel data with manual ROI placement by observer 1.**
(XLSX)

**S3 File. T1 values for interobserver reproducibility assessment; Excel data with semiautomatic ROI placement by observer 1 and 2.**
(XLSX)

**S4 File. T1 values for intraobserver reproducibility assessment; Excel data with semiautomatic ROI placement by observer 1.**
(XLSX)

## Acknowledgments

We thank Dr. Sasaki T. for the helpful software regarding image analysis using MATLAB.

## Author Contributions

**Conceptualization:** Atsutaka Okizaki.

**Data curation:** Takahiro Uno.

**Formal analysis:** Wakako Kubota.

**Methodology:** Atsutaka Okizaki.

**Project administration:** Shunta Ishitoya.

**Supervision:** Atsutaka Okizaki.

**Visualization:** Eriko Takabayashi, Akiko Oya.

**Writing – original draft:** Sadahiro Nakagawa.

**Writing – review & editing:** Atsutaka Okizaki.

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
