## [Decision Letter · Decision Letter 0]

14 Mar 2023

PONE-D-22-17380Inter- and Intra-rater reproducibility of quantitative T1 measurement using semiautomatic ROI placement in myometriumPLOS ONE

Dear Dr. Nakagawa,

Thank you for submitting your manuscript to PLOS ONE. After careful consideration, we feel that it has merit but does not fully meet PLOS ONE’s publication criteria as it currently stands. Therefore, we invite you to submit a revised version of the manuscript that addresses the points raised during the review process.

We look forward to receiving your revised manuscript.

Kind regards,

Parasuraman Padmanabhan, Ph.D

Academic Editor

PLOS ONE

Journal Requirements:

2. Please note that PLOS ONE has specific guidelines on code sharing for submissions in which author-generated code underpins the findings in the manuscript. In these cases, all author-generated code must be made available without restrictions upon publication of the work. 

Please review our guidelines at https://journals.plos.org/plosone/s/materials-and-software-sharing#loc-sharing-code and ensure that your code is shared in a way that follows best practice and facilitates reproducibility and reuse.

"Atsutaka Okizaki received Research funding of Nihon Medi-physics Co. and FUJIFILM Toyama Chemical Co., Ltd. The other authors have no conflict of interest."

4. Thank you for stating the following in the Financial Disclosure section:  

We note that you received funding from a commercial source: Nihon Medi-physics Co. and UJIFILM Toyama Chemical Co., Ltd.

Within this Competing Interests Statement, please confirm that this does not alter your adherence to all PLOS ONE policies on sharing data and materials by including the following statement: ""This does not alter our adherence to PLOS ONE policies on sharing data and materials.” (as detailed online in our guide for authors http://journals.plos.org/plosone/s/competing-interests).  

If there are restrictions on sharing of data and/or materials, please state these. Please note that we cannot proceed with consideration of your article until this information has been declared. 

7. Please upload a new copy of Figure 1 as the detail is not clear. Please follow the link for more information:

https://blogs.plos.org/plos/2019/06/looking-good-tips-for-creating-your-plos-figures-graphics/

https://blogs.plos.org/plos/2019/06/looking-good-tips-for-creating-your-plos-figures-graphics/

**Additional Editor Comments:**

This paper need more editing in terms of language and other MRI related data

The image quality are very poor and high resolution image would be helpful

discussion need to be elobarated with additional references.

Reviewers' comments:

Reviewer's Responses to Questions

**Comments to the Author**

1. Is the manuscript technically sound, and do the data support the conclusions?

Reviewer #1: Yes

2. Has the statistical analysis been performed appropriately and rigorously? 

Reviewer #1: Yes

3. Have the authors made all data underlying the findings in their manuscript fully available?

Reviewer #1: Yes

4. Is the manuscript presented in an intelligible fashion and written in standard English?

Reviewer #1: Yes

5. Review Comments to the Author

Reviewer #1: 1. Manual ROI placement was performed using MATLAB: Please specify which method was used

2. In-house software using Microsoft Visual Studio 2010: many semi automatic segmentation models are there, please specify which algorithm was employed in this research work, upon observation of outputs, seems to have developed an active contour model for ROI extraction, if so, please specify in detail about that.

3. Any preprocessing was done prior to ROI placement

4. Quantitative T1 (qT1) values with variable flip angles (VFA) are important for characterizing lesions in 24 patients with pelvic disease: whether this is applicable only for pelvic disease diagnosis

5. Please improve the quality of figure 3

6. PLOS authors have the option to publish the peer review history of their article (what does this mean?). If published, this will include your full peer review and any attached files.

Reviewer #1: **Yes: **Dr S.N Kumar

---

## [Author Response · Author response to Decision Letter 0]

23 Apr 2023

Replay to the reviewer and editor

23/April/2023

Dear Sir: Editor-In-Chief

Manuscript reference No. PONE-D-22-17380

Please find attached a revised version of our manuscript “Inter- and Intra-rater reproducibility of quantitative T1 measurement using semiautomatic ROI placement in myometrium”, which we would like to resubmit for publication in PLOS ONE.

Thank you much for your reviews. Your comments were highly insightful and enabled us greatly improve the quality of our manuscript. In the following pages are our point-by-point responses to each of the comments. Furthermore, we attached changed parts of our manuscript. We revised those parts according to the editor’s and reviewer’s comments as 'Response to Reviewers'.

Revisions in the text are shown using yellow highlight for changes and green highlight for deletion. We hope that the revisions in the manuscript are suitable for publication in PLOS ONE.

We shall look forward to hearing from you.

Response to Reviewers

Associate editor:

Reply>

Thank you for your kind and informative comments on our manuscript. We greatly appreciate your evaluation of our work. As per your suggestion, we have revised our cover letter to include our updated financial disclosure statement.

Thank you again for your time and consideration. We look forward to hearing from you.

Associate editor:

Reply>

Thank you for your message regarding resubmitting our figure files (Figure 1-3). We have carefully reviewed the reviewer's comments and have made the necessary revisions to our figures to meet the requirements for publication in your journal. We appreciate the guidelines provided at the end of the letter and have ensured that our revised figures meet the specified criteria. Thank you for your continued guidance and support throughout this process.

Associate editor:

Reply>

Thank you for your feedback regarding the depositing of our laboratory protocols to Protocols.io. After careful consideration, we have decided that depositing our protocols may not be the most effective way to enhance the reproducibility of our results. However, we understand the importance of reproducibility and have taken steps to ensure that our results can be reproduced.

To enhance the reproducibility of our results, we have provided detailed information on the scan parameters used in our MRI study, as well as the program code for our semiautomatic ROI placement and the procedure for manual ROI placement. We believe that these measures will enable others to reproduce our results and further the field of research.

Thank you for your valuable feedback, and please let us know if you have any further suggestions.

Associate editor:

Reply>

Thank you for your message regarding the style requirements for our manuscript. We have consulted the PLOS ONE style templates and made the necessary adjustments to ensure that our manuscript and Title page conform to PLOS ONE's guidelines for formatting and file naming. We appreciate your guidance and assistance in ensuring that our manuscript meets the required standards for publication in your journal.

Associate editor:

2. Please note that PLOS ONE has specific guidelines on code sharing for submissions in which author-generated code underpins the findings in the manuscript. In these cases, all author-generated codes must be made available without restrictions upon publication of the work. 

Please review our guidelines at https://journals.plos.org/plosone/s/materials-and-software-sharing#loc-sharing-code and ensure that your code is shared in a way that follows best practices and facilitates reproducibility and reuse.

Reply>

Thank you for bringing to our attention the specific guidelines on code sharing for submissions where author-generated code is fundamental to the manuscript's findings. We have carefully reviewed the guidelines at https://journals.plos.org/plosone/s/materials-and-software-sharing#loc-sharing-code and have ensured that our code is shared in a manner that adheres to best practices, promoting reproducibility and facilitating reuse.

To produce semi-automatic placement of regions of interest (ROIs), we utilized Microsoft Visual Studio 2010 to generate the code. We also developed MATLAB code to generate T1 maps from two SPRG images and created a procedure to convert DICOM files to JPEG format. Additionally, we developed MATLAB code that produces code for the manual placement of ROIs in DICOM format.

In compliance with the guidelines, we have shared the generated code, a test dataset, and the procedure on Zenodo at https://zenodo.org/deposit?page=1&size=20. These resources have been assigned Digital Object Identifiers (DOIs) through Zenodo.

Furthermore, we have included the procedure for replicating our results using these resources in the supporting information (S1-4 Appendix). We have confirmed that these resources are publicly accessible and can be used to replicate the analysis conducted in our study. We hope that other researchers will find these resources valuable in investigating similar research questions.

Thank you for your attention to this matter.

< Materials and Methods P7 line111>

T1 maps were generated pixel by pixel using MATLAB (MathWorks, Natick, MA) with two flip angles (FAs) of 4° and 18° [3]

↓

T1 maps were generated pixel by pixel using MATLAB (MathWorks, Natick, MA) with two flip angles (FAs) of 4° and 18° (S1 Appendix) [3]. The code used to create the T1 map was made available through Zenodo, DOI: 10.5281/zenodo.7808462 (https://doi.org/10.5281/zenodo.7808462).

< Materials and Methods P8 line128>

Manual ROI placement was performed using MATLAB. First, a whole uterus ROI was manually placed to trace the external contour of the myometrium (Fig.1a).

↓

Manual ROI placement was performed using MATLAB. First, a whole uterus ROI was manually placed to trace the external contour of the myometrium (Fig.1a). (S2 Appendix) using the following procedure: (1) The T1 map data was transferred to the MATLAB workspace; (2) An IR image was selected by the observer in the MATLAB application to place the ROI; (3) The provided code was executed to open the image segmentation application in MATLAB; (4) A whole uterus ROI was manually placed to trace the external contour of the myometrium; (5) An endometrial ROI was manually placed to trace the external contour of the endometrium; (6) The myometrial ROI was created by subtracting the endometrial ROI from the whole uterus ROI; (7) Mean qT1 values were calculated from the corresponding pixel values of the T1 map within the myometrial ROI. The code for manual ROI placement was made available through Zenodo, DOI: DOI/10.5281/zenodo.7811343 (https://doi.org/10.5281/zenodo.7811343).

< Materials and Methods P9 line144>

We developed an in-house software using Microsoft Visual Studio 2010 (Microsoft Corporation, Redmond, WA, USA) to read and display digital imaging and communications in medicine (DICOM) files on a personal computer with Windows 10, 64-bit operating system (Microsoft). The myometrial ROI was placed using the following semiautomatic procedure (Fig 1d-f): (1) the center of the endometrium was determined by the user; (2) the user traced the external contour of the myometrium (Fig 1d); (3) the endometrial ROI was automatically extracted by the software using a signal intensity threshold (Fig 1e), which was initially set at within ±3% of the endometrial signal intensity determined by the user and was increased in increments of 1% to maximize the area of the endometrial ROI, while excluding the external contour of the myometrium; (4) pixel values were sampled from the area between the endometrial ROI and external contour of the myometrium; (5) the mean and standard deviation of the sampled pixel values were calculated; (6) the myometrial ROI was extracted automatically based on pixels that were within ±1.96 standard deviations of the mean signal intensity; (7) the myometrial ROI was placed on the SPGR image; (8) we calculated the mean pixel value of the myometrial ROI; (9) the sum of squared deviation was calculated with variation of ROI locations, and the vertical and horizontal coordinates were changed from −10 to +10 pixels; (10) the optimal combination of vertical and horizontal coordinates with minimum sum of squared deviation was obtained, and the appropriateness of the ROI location was visually confirmed by experts; and (11) the mean qT1 value was calculated from the corresponding pixel values of the T1 map within the optimally placed myometrial ROI (Fig 1f). Eventually, the required operations by the user were only 2 steps: (1) click center of endometrium, (2) roughly draw the shape of myometrium manually.

↓

We developed an in-house software using Microsoft Visual Studio 2010 (Microsoft Corporation, Redmond, WA, USA) to read and display digital imaging and communications in medicine (DICOM) files on a personal computer with Windows 10, a 64-bit operating system (Microsoft). Prior to analysis, we converted the DICOM images to JPEG format using MATLAB (S3 Appendix). The source code for the in-house software used to read and display DICOM files, as well as the MATLAB code used to convert DICOM images to JPEG format, has been made available through Zenodo, DOI: 10.5281/zenodo.7807266 (https://doi.org/10.5281/zenodo.7807266).

The myometrial ROI was placed using the following semiautomatic procedure (Fig 1d-f, S4 Appendix): (1) the center of the endometrium was determined by the user; (2) the user traced the external contour of the myometrium (Fig 1d); (3) the endometrial ROI was automatically extracted by the software using a signal intensity threshold (Fig 1e), which was initially set at within ±3% of the endometrial signal intensity determined by the user and was increased in increments of 1% to maximize the area of the endometrial ROI, while excluding the external contour of the myometrium; (4) pixel values were sampled from the area between the endometrial ROI and external contour of the myometrium; (5) the mean and standard deviation of the sampled pixel values were calculated; (6) the myometrial ROI was extracted automatically based on pixels that were within ±1.96 standard deviations of the mean signal intensity; (7) the myometrial ROI was placed on the SPGR image; (8) we calculated the mean pixel value of the myometrial ROI; (9) the sum of squared deviation was calculated with variation of ROI locations, and the vertical and horizontal coordinates were changed from −10 to +10 pixels; (10) the optimal combination of vertical and horizontal coordinates with minimum sum of squared deviation was obtained, and the appropriateness of the ROI location was visually confirmed by experts; and (11) the mean qT1 value was calculated from the corresponding pixel values of the T1 map within the optimally placed myometrial ROI (Fig 1f). Eventually, the required operations by the user were only 2 steps: (1) click the center of the endometrium, (2) roughly draw the shape of the myometrium manually. Our algorithm did not use any active contour model for ROI extraction. The Code for semiautomatic ROI placement was made available through the Zenodo, DOI: DOI/10.5281/zenodo.7855051 (https://doi.org/10.5281/zenodo.7855051).

Associate editor:

"Atsutaka Okizaki received Research funding of Nihon Medi-physics Co. and FUJIFILM Toyama Chemical Co., Ltd. The other authors have no conflict of interest."

Replay>

Thank you very much for providing us with such detailed feedback. We truly appreciate it. We want to let you know that we have carefully considered your comments, and as the funders did not play a role in this study, we have revised our statement on competing interests accordingly. The updated statement now reads as follows: "The funders had no involvement in the study design, data collection and analysis, decision to publish, or preparation of the manuscript." We have made sure to include this revised statement in our cover letter.

Associate editor:

4. Thank you for stating the following in the Financial Disclosure section: 

We note that you received funding from a commercial source: Nihon Medi-physics Co. and FUJIFILM Toyama Chemical Co., Ltd.

Within this Competing Interests Statement, please confirm that this does not alter your adherence to all PLOS ONE policies on sharing data and materials by including the following statement: “This does not alter our adherence to PLOS ONE policies on sharing data and materials.” (As detailed online in our guide for authors http://journals.plos.org/plosone/s/competing-interests). 

If there are restrictions on sharing of data and/or materials, please state these. Please note that we cannot proceed with consideration of your article until this information has been declared. 

Replay>

Thank you very much for your valuable feedback. We would like to provide an amended Competing Interests Statement that explicitly states the commercial funders, Nihon Medi-physics Co. and FUJIFILM Toyama Chemical Co., Ltd.: “This does not alter our adherence to PLOS ONE policy on sharing data and materials”.

We confirmed that this statement does not affect our adherence to all PLOS ONE policies on sharing data and materials. Please let us know if any further amendments are necessary.

Associate editor:

Replay>

Thank you very much for your helpful and informative feedback. The minimal dataset underlying the results described in this study, including the inter- and intra-observer reliability of qT1 evaluation times obtained by observers 1 and 2 using manual and semiautomatic ROI placements, is provided in the manuscript and its Supporting Information files (S5–8 Files). We have created a Data Availability Statement and included it in our cover letter, which reads as follows:

Data Availability Statement:

The minimal dataset underlying the results described in this study are within the manuscript and its Supporting Information files. The source code analysis of Microsoft Visual Studio 2010 image analysis code and MATLAB, along with an example image that supports the findings of this study, has been openly published on Zenodo at [https://zenodo.org/deposit?page=1&size=20]. Additionally, any other data required to fully replicate the reported study findings can be obtained by contacting the corresponding author upon reasonable request.

Thank you for your guidance in ensuring our manuscript meets PLOS data policy requirements. Please let us know if any further revisions or clarifications are needed.

Associate editor:

Replay>

Thank you very much in advance for your kind support.

We have provided the ORCID iD for the corresponding author, Atsutaka Okizaki, below:

ORCID iD: [0000-0001-8203-827X]

Associate editor:

7. Please upload a new copy of Figure 1 as the detail is not clear. Please follow the link for more information:

https://blogs.plos.org/plos/2019/06/looking-good-tips-for-creating-your-plos-figures-graphics/

https://blogs.plos.org/plos/2019/06/looking-good-tips-for-creating-your-plos-figures-graphics/

Replay>

Thank you very much for your helpful and informative comment. We have corrected the resolution of Figure 1 based on your comment and have attached a clearer version of Figure 1 for your convenience.

Reviewer #1: 

1. Manual ROI placement was performed using MATLAB: Please specify which method was used.

Reply>

Thank you for your kind and informative comments on our manuscript. We have developed MATLAB code that generates code for the manual placement of ROIs in DICOM format. To comply with the guidelines, we have shared the generated code, a test dataset, and the procedure on Zenodo at DOI: 10.5281/zenodo.7811343. The resource has been assigned DOI through Zenodo. Additionally, we have included the procedure for replicating our results using these resources in the supporting information (S2 Appendix).

< Materials and Methods P8 line128>

Manual ROI placement was performed using MATLAB. First, a whole uterus ROI was manually placed to trace the external contour of the myometrium (Fig.1a).

↓

Manual ROI placement was performed using MATLAB. First, a whole uterus ROI was manually placed to trace the external contour of the myometrium (Fig.1a). (S2 Appendix) using the following procedure: (1) The T1 map data was transferred to the MATLAB workspace; (2) An IR image was selected by the observer in the MATLAB application to place the ROI; (3) The provided code was executed to open the image segmentation application in MATLAB; (4) A whole uterus ROI was manually placed to trace the external contour of the myometrium; (5) An endometrial ROI was manually placed to trace the external contour of the endometrium; (6) The myometrial ROI was created by subtracting the endometrial ROI from the whole uterus ROI; (7) Mean qT1 values were calculated from the corresponding pixel values of the T1 map within the myometrial ROI. The code for manual ROI placement was made available through Zenodo, DOI: DOI/10.5281/zenodo.7811343 (https://doi.org/10.5281/zenodo.7811343).

Reviewer #1: 

2. In-house software using Microsoft Visual Studio 2010: many semi automatic segmentation models are there, please specify which algorithm was employed in this research work, upon observation of outputs, seems to have developed an active contour model for ROI extraction, if so, please specify in detail about that.

Replay>

Thank you for bringing this to our attention. Our algorithm does not utilize any active contour model for ROI extraction. Instead, we fully developed and programmed a semi-automatic ROI placement approach from scratch. The program code has been published on Zenodo: DOI: 10.5281/zenodo.78550551.

< Materials and Methods P9 line160>

Eventually, the required operations by the user were only 2 steps: (1) click center of endometrium, (2) roughly draw the shape of myometrium manually.

↓

Eventually, the required operations by the user were only 2 steps: (1) click center of endometrium, (2) roughly draw the shape of myometrium manually. Our algorithm did not use any active contour model for ROI extraction.

Reviewer #1: 

3. Any preprocessing was done prior to ROI placement

Replay>

Thank you for your message. I'm glad to have been able to assist you.

To further clarify, prior to running the semiautomatic analysis, the authors developed a MATLAB code (available on Zenodo, DOI: 10.5281/zenodo.7808462) for generating T1 maps from two SPRG images. Additionally, MATLAB (available on Zenodo, DOI: 10.5281/zenodo.7807266) was used by the authors to convert the DICOM images into JPEG format, with automatic optimization of brightness during the conversion process.

Reviewer #1: 

4. Quantitative T1 (qT1) values with variable flip angles (VFA) are important for characterizing lesions in 24 patients with pelvic disease: whether this is applicable only for pelvic disease diagnosis

Replay>

Thank you for your inquiry regarding the applicability of our software beyond the uterus. While our software was originally designed for the analysis of the uterus, we believe that its programmatic design may potentially allow for its use in analyzing other organs as well, including the pelvic region. However, we acknowledge that further research is necessary to confirm this hypothesis. We are committed to continuing our investigation and development of this software to explore its potential for expanding its applications. 

< Discussion P14 line234>

The algorithm for semiautomatic ROI placement provided high reproducibility. In principle, the algorithm can be applied to other medical imaging modalities for delineating the uterus because it uses pixel values that correspond with anatomical structures.

↓

The algorithm for semiautomatic ROI placement provided high reproducibility. In principle, the algorithm can be applied to other medical imaging modalities for delineating the uterus because it uses pixel values that correspond with anatomical structures. Although this software was specifically designed for the analysis of the uterus, we believe that its programmatic design theoretically allows for its use in analyzing other organs as well. With further research, it may be possible to extend the use of this software beyond its current scope. Therefore, we plan to continue investigating and developing this software to explore its potential for expanding applications.

Reviewer #1: 

5. Please improve the quality of figure 3

Replay>

Thank you for your message. We greatly appreciate your helpful and informative comment. As per your suggestion, we have corrected the resolution of Figure 3 and have also attached a clearer version of Figure 3 for your convenience.

---

## [Decision Letter · Decision Letter 1]

27 Nov 2023

PONE-D-22-17380R1Inter- and Intra-rater reproducibility of quantitative T1 measurement using semiautomatic ROI placement in myometriumPLOS ONE

Dear Dr. Nakagawa,

Thank you for submitting your manuscript to PLOS ONE. After careful consideration, we feel that it has merit but does not fully meet PLOS ONE’s publication criteria as it currently stands. Therefore, we invite you to submit a revised version of the manuscript that addresses the points raised during the review process.

We look forward to receiving your revised manuscript.

Kind regards,

Lorenzo Faggioni, M.D., Ph.D.

Academic Editor

PLOS ONE

Journal Requirements:

Reviewers' comments:

Reviewer's Responses to Questions

**Comments to the Author**

1. If the authors have adequately addressed your comments raised in a previous round of review and you feel that this manuscript is now acceptable for publication, you may indicate that here to bypass the “Comments to the Author” section, enter your conflict of interest statement in the “Confidential to Editor” section, and submit your "Accept" recommendation.

Reviewer #2: (No Response)

2. Is the manuscript technically sound, and do the data support the conclusions?

Reviewer #2: Yes

3. Has the statistical analysis been performed appropriately and rigorously? 

Reviewer #2: Yes

4. Have the authors made all data underlying the findings in their manuscript fully available?

Reviewer #2: Yes

5. Is the manuscript presented in an intelligible fashion and written in standard English?

Reviewer #2: No

6. Review Comments to the Author

Reviewer #2: Dear authors,

It was a pleasure to peer-review this paper investigating the inter- and intra-rater reproducibility of manual and semi-automatic quantitative T1 measurement on myometrium in healthy volunteers.

Some minor issues should be addressed:

1. The abstract should be rewritten to provide essential information to the readers. The purpose heading is quite long, providing some unnecessary information, while the material and methods one is quite short, without any information regarding the statistics analysis. Be sure to define all abbreviations (e.g. IR). Purpose and conclusions should be aligned, in the conclusion heading manual segmentation is not mentioned at all.

2. Introduction: The manuscript requires a careful proof-reading, preferably by an English native speaker, many errors and typos (Line 80, pervious) are present, especially in this heading.

Some relevant references are missing (e.g. Line 70, Line 90).

Please define all the abbreviations (e.g. UAE, IR).

Lines 81-82 Please rephrase

Line 105, What does “continuity” means in this setting?

3. Provide a clear statement regarding study nature

4. Line 148, “There were no adequate”. Maybe is it “inadequate”? Otherwise I’m not getting what this sentence mean.

5. Results: Consider to add more results (and less cross-reference to tables or figures) and p-values in this sections.

6. How many patients were excluded due to the presence of mass lesions? It is stated in the methods the presence of this exclusion criterion.

7 Discussion, especially after the particularly long introduction, sounds a bit redundant. Consider to revise both these sections to avoid redundancy, trying to focus the discussion on drawing conclusions regarding the comparison between manual and semiautomatic segmentation.

7. PLOS authors have the option to publish the peer review history of their article (what does this mean?). If published, this will include your full peer review and any attached files.

Reviewer #2: **Yes: **Salvatore Claudio Fanni

---

## [Author Response · Author response to Decision Letter 1]

1 Jan 2024

Replay to the reviewer and editor

30/December/2023

Dear Sir: Editor-In-Chief

Manuscript reference No. PONE-D-22-17380R1

We are pleased to resubmit our revised manuscript, “Inter- and intra-rater reproducibility of quantitative T1 measurement using semiautomatic region of interest placement in myometrium”, for your consideration in PLOS ONE.

Thank you very much for your reviews. Your comments were highly insightful and enabled us to improve the quality of our manuscript greatly. In the following pages are our point-by-point responses to each of the comments. Furthermore, we attached changed parts of our manuscript. We revised those parts according to the editor’s and reviewer’s comments as ‘Response to Reviewers'.

Revisions in the text are shown using yellow highlight for changes and green highlight for deletion. We hope that the revisions in the manuscript are suitable for publication in PLOS ONE.

We shall look forward to hearing from you.

Response to Reviewers

Associate editor:

Reply>

Thank you for your kind and informative comments on our manuscript. We greatly appreciate your evaluation of our work. As per your suggestion, we submitted our laboratory protocols for qT1 measurements to protocols.io, aiming to improve the reproducibility of our results. The uploaded files were S1-4 Appendix of program code for qT1 measurements. 

T1 maps were generated pixel by pixel using MATLAB (MathWorks, Natick, MA) with two flip angles (FAs) of 4° and 18° (S1 Appendix). Manual ROI placement was performed using MATLAB (S2 Appendix). Before analysis, we converted the DICOM images to JPEG format using MATLAB (S3 Appendix). The myometrial ROI was placed using the following semiautomatic procedure (S4 Appendix).

Thank you again for your time and consideration. We look forward to hearing from you.

Associate editor:

Reply>

We have carefully reviewed the reviewer's comments and made the necessary revisions to our manuscript. In response to your specific request, we have revised the reference list as follows:

The deleted reference numbers are marked in green:

7, 8, 15, 18, 21, 22

The changed reference numbers are marked in yellow:

9 →　7

10 →　8

11 →　9

12→　10

13 →　11

14 →　12

16→　13 

17 →　14

19 →　15

20 →　16

23 →　17

24 →　18

25 →　19

26 →　20

Associate editor:

Reply>

We have resubmitted our figure files that were uploaded to the PACE digital diagnostic tool and downloaded from PACE.

Reviewer #2: 

1. The abstract should be rewritten to provide essential information to the readers. The purpose heading is quite long, providing some unnecessary information, while the material and methods one is quite short, without any information regarding the statistics analysis. Be sure to define all abbreviations (e.g. IR). Purpose and conclusions should be aligned, in the conclusion heading manual segmentation is not mentioned at all.

Reply>

Thank you for your message regarding resubmitting our abstract. In response to your specific comments, we have revised the purpose to eliminate unnecessary details, and we have expanded the material and methods section to include information about the statistical analysis. Furthermore, we have ensured that all abbreviations, including "IR," are explicitly defined within the abstract. Furthermore, we added the description for the manual ROI placement in the conclusion.

< Abstract P2 line45>

Purpose: Quantitative T1 (qT1) values with variable flip angles (VFA) are important for characterizing lesions in patients with pelvic disease. QT1 values vary depending on the location, shape, and area of the region of interest (ROI) when the ROI is set manually.

The objective of this study was to investigate the inter- and intraobserver reproducibility of qT1 measurements using manual and semiautomatic ROI settings. We hypothesized that the semiautomatic method, which uses a 3D anatomical relationship between the myometrium and other tissues, may be useful for minimizing the variation in ROI settings and thus, improve qT1 reproducibility.

↓

Purpose: Quantitative T1 (qT1) values with variable flip angles (VFA) are important for characterizing lesions in patients with pelvic disease. QT1 values vary depending on the location, shape, and area of the region of interest (ROI) when the ROI is set manually.

The objective of this This study was aimed to investigate the inter- and intraobserver reproducibility of quantitative T1 (qT1) measurements using manual and semiautomatic region of interest (ROI) settings placements. We hypothesized that the usefulness of the semiautomatic method, which uses utilizes a 3D three-dimensional (3D) anatomical relationship between the myometrium and other tissues, may be useful for minimizing the ROI placement variation, thereby improving in ROI settings and thus, improve qT1 reproducibility compared to the manual approach. The semiautomatic approach, which considered anatomical relationships, was expected to enhance reproducibility by reducing ROI placement variabilities.

< Abstract P2 line52>

Materials and methods: Twenty-three healthy female volunteers were investigated. The data with VFA and IR were acquired using 3D-spoiled gradient echo and spin echo sequences, respectively. T1 maps were generated with VFA. Manual and semiautomatic ROI placements were performed independently. Mean qT1 values were calculated from the T1 maps using the corresponding pixel values of the myometrial ROI. Inter- and intraobserver reproducibility of qT1 values were assessed.

↓

Materials and methods: Twenty-three This study recruited 23 healthy female volunteers were investigated. The data. Data with variable flip angle (VFA) and IR inversion recovery were acquired using 3D-spoiled gradient echo and spin echo sequences, respectively. T1 maps were generated with VFA. Manual and semiautomatic ROI placements were performed independently conducted. Mean qT1 values were calculated from the T1 maps using the corresponding pixel values of the myometrial ROI. Inter- and intraobserver reproducibility of qT1 values were assessed was investigated. The inter- and intraobserver reproducibility of qT1 values was evaluated by calculating the coefficient of variation (CoV). Further, reproducibility was evaluated with inter- and intraobserver errors and intraclass correlation coefficients (ICCs). Bland–Altman analysis was utilized to compare the results, estimate bias, and determine the limits of agreement.

< Abstract P2 line57>

The mean inter- and intraobserver error for semiautomatic ROI placement were also significantly lower than those for manual ROI placement (p < 0.05 and p < 0.01, respectively).

↓

The Further, the mean inter- and intraobserver error errors for semiautomatic ROI placement were also significantly lower than those for manual ROI placement (p < 0.05 and p < 0.01, respectively).

< Abstract P2 line62>

Conclusion: Semiautomatic ROI placement has high reproducibility of qT1 measurements.

↓

Conclusion: Semiautomatic ROI placement has demonstrated high reproducibility of qT1 measurements compared with manual methods. Semiautomatic ROI placement may be useful for evaluating uterine qT1 with high reproducibility.

Reviewer #2: 

2. Introduction: The manuscript requires a careful proof-reading, preferably by an English native speaker, many errors and typos (Line 80, pervious) are present, especially in this heading.

Replay>

Our revised manuscript was edited by a native English speaker through a new editing process.

We have deleted this sentence concluding type error (Line 80, pervious).

The manuscript has undergone additional English proofreading, and we sent a new editorial certificate. We have reviewed and revised the title through English proofreading.

< Introduction P4 line80>

Pervious paper showed peak of qT1 in the uterine fibroid was markedly decreased after the therapy, although that in the myometrium did not lead to a large change [15]

↓

deleted

< Title >

Inter- and Intra-rater reproducibility of quantitative T1 measurement using semiautomatic ROI placement in myometrium

↓

Inter- and Intra intra-rater reproducibility of quantitative T1 measurement using semiautomatic ROI region of interest placement in myometrium

Reviewer #2: 

Some relevant references are missing (e.g. Line 70, Line 90).

Replay>

Thank you for your message. We have added relevant references to the sentence in Line 70, which removed the text including the relevant references to the sentence in Line 90.

< Introduction P4 line69>

Quantitative image analysis has the advantage of better tissue identification and classification compared with visual assessment of traditional T1- and T2-weighted images.

↓

Quantitative image analysis has the advantage of better tissue identification and classification compared with the visual assessment of traditional T1- and T2-weighted images [4-6].

< Introduction P4 line89>

Of these pulse sequences, IR is most frequently used as the gold standard for qT1 evaluation.

↓

deleted

Reviewer #2: 

Please define all the abbreviations (e.g. UAE, IR).

Replay>

We have defined the abbreviations for the words that were pointed out. We have checked all the abbreviations.

<Introduction P5 line88>

Various pulse sequences, including IR [17], modified look-locker inversion recovery (MOLLI) [2], and VFA [18], have been used for measuring qT1.

↓

deleted

We have added the below text in the ‘Subjects’ of the 'Materials and Methods' section.

< Materials and Methods P6 line114>

none

↓

This study excluded subjects who met the following exclusion criteria: (a) subjects who presented with uterine mass lesions (n = 5), (b) subjects who had misregister between inversion recovery (IR) and each VFA image (n = 13), and (c) subjects who observed motion artifacts in the spoiled gradient recalled echo (SPGR) image acquired for qT1 with VFA (n = 15). Our study enrolled 23 healthy women volunteers (age range: 22–51 years; mean age ± standard deviation [SD]: 33 ± 11 years).

<Introduction P4 line78>

Pervious paper showed peak of qT1 in the uterine fibroid was markedly decreased after the therapy, although that in the myometrium did not lead to a large change [15]. The assessment of myometrium may desire for the high intensity focused ultrasound treatment.

↓

Pervious paper showed peak of qT1 in the uterine fibroid was markedly decreased after the therapy, although that in the myometrium did not lead to a large change [15]. The assessment of myometrium may desire for the high intensity focused ultrasound treatment.

 Dominant leiomyoma qT1 measures were used to predict uterine and dominant leiomyoma size reductions after uterine artery embolization (UAE) [12].

Reviewer #2: 

Lines 81-82 Please rephrase

Replay>

We greatly appreciate your helpful and informative comment. As per your suggestion, we have deleted this text.

<Introduction P4 line79>

The assessment of myometrium may desire for the high intensity focused ultrasound treatment.

↓

deleted

Reviewer #2: 

Line 105, What does “continuity” means in this setting?

Replay>

As per your suggestion, we have corrected the text.

<Introduction P6 line103>

We hypothesized that a semiautomatic method that uses a 3D anatomical relationship between the myometrium and endometrium and continuity of MRI may be useful to minimize variation and improve the reproducibility of the ROI settings.

↓

We hypothesized that a semiautomatic method that uses, using a 3D anatomical relationship between the myometrium and endometrium and continuity of MRI, may be useful to minimize for minimizing variation and improve the by manual ROI placements and improving ROI placement reproducibility of the ROI settings.

Reviewer #2: 

3. Provide a clear statement regarding study nature

Replay>

We have extensively revised the introduction to provide a clear statement regarding the nature of our study. 

<Introduction P6 line103>

A highly reproducible ROI placement without the need for a contrast agent is preferable for quantitative MRI. We hypothesized that a semiautomatic method that uses a 3D anatomical relationship between the myometrium and endometrium and continuity of MRI may be useful to minimize variation and improve the reproducibility of the ROI settings. We developed a semiautomatic myometrial ROI placement method that requires no contrast agent. The objective of this study is to investigate the inter- and intraobserver reproducibility of qT1 measurements using manual and semiautomatic ROI settings.

↓

A highly Highly reproducible ROI placement without the need for a contrast agent is preferable for quantitative MRI. 

Therefore, we developed a semiautomatic myometrial ROI placement method without a contrast agent. We hypothesized that a semiautomatic method that uses, using a 3D anatomical relationship between the myometrium and endometrium and continuity of MRI, may be useful to minimize for minimizing variation and improve the by manual ROI placements and improving ROI placement reproducibility of the ROI settings. We developed a semiautomatic myometrial ROI placement method that requires no contrast agent. The objective of this. This study is aimed to investigate the inter- and intraobserver reproducibility of qT1 measurements using manual and semiautomatic ROI settings placements.

Reviewer #2: 

4. Line 148, “There were no adequate”. Maybe is it “inadequate”? Otherwise I’m not getting what this sentence mean.

Replay>

As per your suggestion, we have removed the text including the sentence.

< Materials and Methods P4 line148>

There were no adequate image quality and image misregistration between IR image and 2 SPGRs on visual assessment.

↓

There were no adequate, evaluated the MR images using IR images and two SPGRs to confirm image quality and image misregistration between IR image registration.

Reviewer #2: 

5. Results: Consider to add more results (and less cross-reference to tables or figures) and p-values in this sections.

Replay>

As per your suggestion, we have added more results and p-values in this section.

< Materials and Methods P24 line252>

A negative proportional bias was observed between the first and second measurements of qT1 using manual ROI placement (Fig.3c). However, no proportional bias was observed using semiautomatic ROI placement (Fig.3d).

The means and standard deviations of time with manual and semiautomatic ROI placements were 55±4 sec, 44±5 sec, respectively. The times with semiautomatic ROI placement were significant smaller than those with manual ROI placement (P < 0.001).

↓

Positive proportional bias emerged in the qT1 measurements between observers 1 and 2 using semiautomatic ROI placement (R = 0.461, p = 0.027; Fig. 3b). A negative proportional bias was observed determined between the first and second qT1 measurements of qT1 using manual ROI placement (R = 0.532, p = 0.009; Fig. 3c). However, no proportional bias was observed using semiautomatic ROI placement (Fig.in the measurements between observers 1 and 2 using manual ROI placement (R = 0.224, p = 0.304; Fig. 3a). Similarly, no proportional bias was observed between the first and second measurements using semiautomatic ROI placement (R = 0.11, p = 0.959; Fig. 3d).

 The means and standard deviations SDs of time with manual and semiautomatic ROI placements were 55 ± 4 sec s and 44 ± 5 sec s, respectively. The times with semiautomatic ROI placement were significant significantly smaller than those with manual ROI placement (Pp < 0.001).

Reviewer #2: 

6. How many patients were excluded due to the presence of mass lesions? It is stated in the methods the presence of this exclusion criterion.

Replay>

We have added the exclusion criteria in the ‘Subjects” section.

< Materials and Methods P6 line112>

Subjects

The study protocol was approved by our institutional research ethics committee at Asahikawa Medical University in accordance with the principles of the Declaration of Helsinki. All subjects were informed regarding the content of the study and were provided written informed consent. Between May 2018 and April 2019, we enrolled 23 healthy women volunteers (age range, 22–51 years; mean age ± standard deviation, 33 ± 11 years) in our study.

↓

The study protocol was approved by our Our institutional research ethics committee at Asahikawa Medical University in accordance with approved the study protocol following the principles of the Declaration of Helsinki. All subjects were informed regarding the content of the study and were provided signed written informed consent. Between From May 2018 and to April 2019, we enrolled 23 healthy women volunteers (age range, 22–51 years; mean age ± standard deviation, 33 ± 11 years)56 subjects underwent pelvic MR imaging, incorporating VFA for qT1 measurements in our study. hospital.

This study excluded subjects who met the following exclusion criteria: (a) subjects who presented with uterine mass lesions (n = 5), (b) subjects who had misregister between inversion recovery (IR) and each VFA image (n = 13), and (c) subjects who observed motion artifacts in the spoiled gradient recalled echo (SPGR) image acquired for qT1 with VFA (n = 15). Our study enrolled 23 healthy women volunteers (age range: 22–51 years; mean age ± standard deviation [SD]: 33 ± 11 years).

Reviewer #2: 

7 Discussion, especially after the particularly long introduction, sounds a bit redundant. Consider to revise both these sections to avoid redundancy, trying to focus the discussion on drawing conclusions regarding the comparison between manual and semiautomatic segmentation.

Replay>

As per your suggestion, we have extensively revised the introduction in response to the reviewer's comments. Furthermore, we have added a comparison of manual and semiautomatic ROI placement in the conclusion.

<Introduction P6 line103>

Quantitative image analysis of magnetic resonance imaging (MRI) data may be useful for comparing subjects. Quantitative T1 (qT1), T2, and apparent diffusion coefficient values have been used to investigate various diseases [1-4]. Quantitative image analysis has the advantage of better tissue identification and classification compared with visual assessment of traditional T1- and T2-weighted images. QT1 values have been used extensively to study the central nervous system over the past several decades [5-7]. Visualizing qT1 maps are more useful for identifying deep cerebellar nuclei than traditional T1-weighted images [6]. Recently, qT1 values have also been used in the diagnoses of heart disease, where the detection rates of acute myocardial edema with qT1 were higher than those with T2-weighted images[8]. Quantitative analyses are applicable for diagnoses not only in the central nervous system and myocardium but also in the pelvis [9-12].

Davenport et al. reported that qT1 using dynamic contrast-enhanced MRI with variable flip angle (VFA) in patients with uterine myoma was important for characterizing the lesion and monitoring drug effects [13]. QT1 in myometrium was also available for investigating uterine pathology. The preinterventional qT1 of leiomyoma and uterus also could predict the ability of UAE to reduce the sizes of the leiomyoma and the histological change of uterus [14]. Pervious paper showed peak of qT1 in the uterine fibroid was markedly decreased after the therapy, although that in the myometrium did not lead to a large change [15]. The assessment of myometrium may desire for the high intensity focused ultrasound treatment.

QT1 values vary depending on various factors, such as magnetic field strength, magnet manufacturer, imaging sequence, image acquisition policy, postprocessing algorithm, body size and age of patients, and measurement method of quantitative values [16, 17]. Therefore, quantitative values are not directly comparable between different institutions without the use of certain reference values, and each institution needs to define reference values for standardization. In principle, such reference values would require high reproducibility of measurements.

Various pulse sequences, including IR[17], modified look-locker inversion recovery (MOLLI)[2], and VFA[18], have been used for measuring qT1. Of these pulse sequences, IR is most frequently used as the gold standard for qT1 evaluation. However, acquisition with IR requires considerable time (from 30 minutes to several hours). Acquisitions with MOLLI and VFA are shorter compared with IR, although MOLLI has the disadvantage of data of only one slice being obtained per acquisition, and thus, acquiring many slices is still time-consuming. By contrast, the VFA method requires a shorter acquisition time because multiple slices are acquired per acquisition.

A robust, reproducible, and standardized method is desirable for quantitative measurements. QT1 values sometimes vary, which can depend on location, shape, and area of region of interest (ROI) when the ROI is set manually. Heye et al. reported that a semiautomatic ROI placement improved the reproducibility of pharmacokinetic parameters, but the method is only available using dynamic enhanced MRI [19]. Chapiro et al. highlighted the usefulness of semiautomated 3D tumor segmentation but noted the necessity of acquiring contrast-enhanced T1-weighted images [20].

Gadolinium contrast agents are commonly used in clinical practice. However, they have been known to sometimes cause side effects, making them unsuitable for patients with severe kidney dysfunction due to avoid nephrogenic systemic fibrosis [21, 22]. Therefore, contrast agents should only be used if necessary.

↓

Quantitative image analysis of magnetic Magnetic resonance imaging (MRI) data quantitative image analysis may be useful for comparing subjects. Quantitative T1 (qT1), T2, and apparent diffusion coefficient values have been used utilized to investigate various diseases [1-4]. Quantitative image analysis has the advantage of better tissue identification and classification compared with the visual assessment of traditional T1- and T2-weighted images [4-6]. QT1 values have been used extensively used to study the central nervous system over the past several decades [5-7]. Visualizing qT1 maps are more useful for identifying deep cerebellar nuclei than traditional T1-weighted images [6]. Recently, qT1 values have also been used in the diagnoses of heart disease, where the detection rates of acute myocardial edema with qT1 were higher than those with T2-weighted images[8]. Quantitative analyses are applicable for diagnoses not only in the central nervous system and myocardium but also in the pelvis[9-12] [7-10].

Davenport et al. reported that revealed qT1, using dynamic contrast-enhanced MRI with variable flip angle (VFA) in patients with uterine myoma was important, to be crucial for characterizing the lesion and monitoring drug effects [1311]. QT1 in myometrium was also available for investigating uterine pathology. The preinterventional qT1 of leiomyoma and uterus also could predict the ability of UAE to reduce the sizes of the leiomyoma and the histological change of uterus[14]. Pervious paper showed peak of qT1 in the uterine fibroid was markedly decreased after the therapy, although that in the myometrium did not lead to a large change[15]. The assessment of myometrium may desire for the high intensity focused ultrasound treatment.

. Dominant leiomyoma qT1 measures were used to predict uterine and dominant leiomyoma size reductions after uterine artery embolization (UAE) [12].

QT1 values vary depending based on various factors, such as including magnetic field strength, magnet manufacturer, imaging sequence, image acquisition policy, postprocessing algorithm, body size and age of patients, and measurement method of quantitative values and patient characteristics [16, 1713, 14]. Therefore, quantitative values are not directly comparable between different institutions without the use of certain reference values, and each institution needs to must define reference values for standardization. In principle, such these reference values would require high reproducibility of for quantitative measurements.

Various pulse sequences, including IR[17], modified look-locker inversion recovery (MOLLI)[2], and VFA[18], have been used for measuring qT1. Of these pulse sequences, IR is most frequently used as the gold standard for qT1 evaluation. However, acquisition with IR requires considerable time (from 30 minutes to several hours). Acquisitions with MOLLI and VFA are shorter compared with IR, although MOLLI has the disadvantage of data of only one slice being obtained per acquisition, and thus, acquiring many slices is still time-consuming. By contrast, the VFA method requires a shorter acquisition time because multiple slices are acquired per acquisition.

A robust, reproducible, and standardized method is desirable for quantitative measurements. QT1 values sometimes vary, which can depend based on the location, shape, and area of the region of interest (ROI) when the ROI is set manually set. Heye et al. reported revealed that a semiautomatic ROI placement improved enhanced the reproducibility of pharmacokinetic parameters, but; however, the method is only available using dynamic enhanced MRI [1915]. Chapiro et al. highlighted emphasized the usefulness of semiautomated three-dimensional (3D) tumor segmentation but noted the necessity of acquiring contrast-enhanced T1-weighted images [2016].

Gadolinium contrast agents are commonly used in clinical practice. However, they have been known to sometimes cause side effects, making them unsuitable for patients with severe kidney dysfunction due to avoid nephrogenic systemic fibrosis[21, 22]. Therefore, contrast agents should only be used if necessary.

A highly Highly reproducible ROI placement without the need for a contrast agent is preferable for quantitative MRI. 

Therefore, we developed a semiautomatic myometrial ROI placement method without a contrast agent. We hypothesized that a semiautomatic method that uses, using a 3D anatomical relationship between the myometrium and endometrium and continuity of MRI, may be useful to minimize for minimizing variation and improve the by manual ROI placements and improving ROI placement reproducibility of the ROI settings. We developed a semiautomatic myometrial ROI placement method that requires no contrast agent. The objective of this. This study is aimed to investigate the inter- and intraobserver reproducibility of qT1 measurements using manual and semiautomatic ROI settings placements.

<Discussion P17 line305>

In conclusion, the semiautomatic ROI placement algorithm had high reproducibility of qT1 measurements.

↓

In conclusion, the semiautomatic ROI placement algorithm had demonstrated high reproducibility of qT1 measurements compared with manual methods. Semiautomatic ROI placement may be useful for evaluating uterine qT1 with high reproducibility.

---

## [Decision Letter · Decision Letter 2]

5 Jan 2024

Inter- and intra-rater reproducibility of quantitative T1 measurement using semiautomatic region of interest placement in myometrium

PONE-D-22-17380R2

Dear Dr. Nakagawa,

We’re pleased to inform you that your manuscript has been judged scientifically suitable for publication and will be formally accepted for publication once it meets all outstanding technical requirements.

Kind regards,

Lorenzo Faggioni, M.D., Ph.D.

Academic Editor

PLOS ONE

Reviewers' comments:

Reviewer's Responses to Questions

**Comments to the Author**

1. If the authors have adequately addressed your comments raised in a previous round of review and you feel that this manuscript is now acceptable for publication, you may indicate that here to bypass the “Comments to the Author” section, enter your conflict of interest statement in the “Confidential to Editor” section, and submit your "Accept" recommendation.

Reviewer #2: All comments have been addressed

2. Is the manuscript technically sound, and do the data support the conclusions?

Reviewer #2: Yes

3. Has the statistical analysis been performed appropriately and rigorously? 

Reviewer #2: Yes

4. Have the authors made all data underlying the findings in their manuscript fully available?

Reviewer #2: Yes

5. Is the manuscript presented in an intelligible fashion and written in standard English?

Reviewer #2: Yes

6. Review Comments to the Author

Reviewer #2: (No Response)

7. PLOS authors have the option to publish the peer review history of their article (what does this mean?). If published, this will include your full peer review and any attached files.

Reviewer #2: No

---

## [Editor Report · Acceptance letter]

18 Jan 2024

PONE-D-22-17380R2 

PLOS ONE

Dear Dr. Nakagawa, 

I'm pleased to inform you that your manuscript has been deemed suitable for publication in PLOS ONE. Congratulations! Your manuscript is now being handed over to our production team.

Kind regards, 

on behalf of

Dr. Lorenzo Faggioni 

Academic Editor

PLOS ONE